# Rescoring and Linearly Combining: A Highly Effective Consensus Strategy for Virtual Screening Campaigns

**DOI:** 10.3390/ijms20092060

**Published:** 2019-04-26

**Authors:** Alessandro Pedretti, Angelica Mazzolari, Silvia Gervasoni, Giulio Vistoli

**Affiliations:** Dipartimento di Scienze Farmaceutiche, Università degli Studi di Milano, Via Mangiagalli, 25, I-20133 Milano, Italy; angelica.mazzolari@unimi.it (A.M.); silvia.gervasoni@unimi.it (S.G.); giulio.vistoli@unimi.it (G.V.)

**Keywords:** enrichment factor, virtual screening, molecular docking, rescore, consensus strategy

## Abstract

The study proposes a novel consensus strategy based on linear combinations of different docking scores to be used in the evaluation of virtual screening campaigns. The consensus models are generated by applying the recently proposed Enrichment Factor Optimization (EFO) method, which develops the linear equations by exhaustively combining the available docking scores and by optimizing the resulting enrichment factors. The performances of such a consensus strategy were evaluated by simulating the entire Directory of Useful Decoys (DUD datasets). In detail, the poses were initially generated by the PLANTS docking program and then rescored by ReScore+ with and without the minimization of the complexes. The so calculated scores were then used to generate the mentioned consensus models including two or three different scoring functions. The reliability of the generated models was assessed by a per target validation as performed by default by the EFO approach. The encouraging performances of the here proposed consensus strategy are emphasized by the average increase of the 17% in the Top 1% enrichment factor (EF) values when comparing the single best score with the linear combination of three scores. Specifically, kinases offer a truly convincing demonstration of the efficacy of the here proposed consensus strategy since their Top 1% EF average ranges from 6.4 when using the single best performing primary score to 23.5 when linearly combining scoring functions. The beneficial effects of this consensus approach are clearly noticeable even when considering the entire DUD datasets as evidenced by the area under the curve (AUC) averages revealing a 14% increase when combining three scores. The reached AUC values compare very well with those reported in literature by an extended set of recent benchmarking studies and the three-variable models afford the highest AUC average.

## 1. Introduction

Virtual screening (VS) involves different computational approaches aimed to identify from among huge molecular databases optimized sets of compounds which have the potential to bind a given biological target and which will undergo high throughput screenings (HTS) in order to identify novel hit compounds [1]. Virtual screening studies have thus the primary objective to reduce the number of compounds to be experimentally screened, thus reducing the costs of the HTS campaigns without limiting their probability of success. In other words, they should be able to generate targeted libraries which are enriched in active compounds compared to the initial (very large) databases.

One of the simplest ways to evaluate the performances of a VS study is to measure how much the resulting datasets are enriched in active compounds by analyzing the so-called enrichment factors (EF) [2]. Similarly, a typical exercise for evaluating the reliability of a VS method involves suitably collected databases in which a set of known active molecules toward a given target is dispersed within a large amount of inactive molecules (the so-called decoys, which usually constitute more than 95% of the entire database) and the VS approach is evaluated by considering its capability to recognize the few active molecules. To do this, several optimized databases are freely available and are used as reference benchmarks in VS evaluations [3]. 

Even though VS can also involve ligand-based approaches (e.g., based on pharmacophore mapping [4,5]), docking methods represent a remarkably productive (and informative) way to perform virtual screening (VS) campaigns and to find potential ligands for a given target [6,7]. By considering the relevance of the VS studies, it come as no surprise that many targeted procedures have been recently proposed to enhance the VS performances. These approaches usually combine different methods which can be arranged in parallel, when combining, for example, docking results produced by different docking programs [8] or in series when docking simulations are preceded by ligand-based approaches for initial filtering analyses [9] and/or followed by post-docking simulations with a view to refining the computed complexes [10]. The rationale of combining in parallel several docking approaches is based on the observation that a proper combination of various approaches by suitably designed consensus strategies might synergistically optimize the reliability of the obtained results by maximizing their advantages and minimizing their drawbacks [11]. 

Even though docking results are clearly influenced by the applied search algorithms, scoring functions represent the most critical factor in determining the overall reliability of docking simulations [12]. On these grounds, a convenient strategy to combine more docking procedures while limiting the computational cost can involve rescoring calculations in which the ligand poses are initially generated by using a single and reasonably satisfactory docking program and then utilized to calculate an extended set of docking scores among which the best performing ones are suitably selected and/or combined [13]. 

As cited above, the procedures, in which different docking scores are combined, usually involve consensus strategies, and the choice of the optimal consensus algorithms plays a critical role in determining the overall VS performances [14]. While linear combinations of docking scores are routinely employed in correlative studies to predict the bioactivity of novel derivatives, they were very scantly utilized in VS simulations apart from few studies based on multi-objective optimization algorithm [15] and on principal component analysis [16]. On these grounds, the present study explores the possibility of applying to VS analyses the recently proposed classification algorithm which generates linear combinations of docking scores as selected by the Enrichment Factor Optimization (EFO) algorithm [17]. Such an approach, which was developed to conveniently classify unbalanced datasets, should find successful applications also when analyzing VS campaigns which indeed involves extremely unbalanced databases. In particular, the docking scores are systematically combined to generate classification models including a user-defined number of independent variables. The coefficients of the resulting equations are calculated by applying an optimization algorithm for non-continuous functions, the goal being to optimize the ranking position of the active molecules. Moreover, local minima are evaded by applying a random sampling algorithm which allows a better optimization of the resulting classification models, whose performances are evaluated by a purposely defined quality function. The potential of such a consensus strategy was evaluated by performing VS analyses using all the 40 DUD datasets which were preferred to those of DUD-E for the huge number of benchmarking studies based on the former which allow extensive comparisons of the here reached results with already published studies [18]. The predictive power of the generated models with two or three variables was assessed by a per target validation in which each DUD dataset was repeatedly subdivided into training and test sets and the models were evaluated and selected by considering their average performances when applied to test sets.

## 2. Results and Discussion

### 2.1. Single Variable Models

The EFO approach can also be utilized when developing single variable models to quickly reveal the best performing scores while avoiding systematic analyses. In this case, the EFO algorithm does not involve optimization processes and thus the models and the corresponding EF values can comprise the entire DUD datasets without validation procedures. Here and in the following sections, the analyses will be subdivided into three parts which involve: (a) the primary Piecewise Linear Potential score (PLP) and its normalized values as directly computed by PLANTS; (b) the various scoring functions as computed by ReScore+ without post-docking minimization and (c) after post-docking minimization. 

Table 1 summarizes the average values for the obtained enrichment factors as subdivided according to the target classes for the three mentioned parts of the study. A bird’s eye analysis of the compiled EF values reveals the beneficial role of the rescoring procedures which allow a marked increase of the reached performances especially when comparing the results obtained by the primary score with those afforded by non-minimized docking scores. The enhancements are more pronounced when considering the top of the ranking (i.e., the EF 1% and EF 2% averages showing increases of 53.6% and 45.7%, respectively) and decline when considering larger part of the ranking as exemplified by the EF 20% averages which show an enhancement of 14.6% only.

While being less pronounced, the complex minimization also affords a beneficial contribution to the obtained performances as documented by the overall increase of 65% in the EF 1% averages when comparing the performances reached by rescoring minimized complexes with those obtained when using only the primary scores. When analyzing the specific results for each target class, the three parts reveal similar trends in which kinases are by far the most challenging targets. Serine proteases are the targets the performances of which reveal the greatest beneficial effect from the rescoring calculations while the EF values for nuclear hormone receptors appear to be poorly sensitive to the rescoring effects.

On average, the similar trends observed for each target class regardless of rescoring and/or of complex minimization indicate that the obtained performances are similarly affected by the reliability of the generated poses, suggesting that rescoring analyses, while showing clearly beneficial effects, cannot completely elude the weaknesses of the utilized docking program.

Appendix A report the enrichment factors and the best equations for each target in the three mentioned analyses. Concerning the analyses based on the PLANT scores, Appendix A shows that most equations include normalized scores and only 6 cases out of 40 involve the total non-normalized score values. In detail, most score normalizations (22 out of 34) are based on molecular weight, thus confirming that ligand size plays a crucial biasing effect on the reliability of docking scores. The analyses on rescoring effect without minimization (Appendix A) confirms the efficacy of the PLANTS scoring functions which are included in 17 equations. The other three groups of scoring functions (namely, the interaction energies calculated by AMMP program [19], the X-Score values [20] and the energy components as computed by VEGA [21]) reveal a comparable occurrence in the remaining 23 cases. Finally, Appendix A reveals that the rescoring after minimization has a significant impact on the occurrence of the included scores as witnessed by the decrease of the PLANTS-based equations (13 vs. 17) and the marked increase of the VEGA-based models (19 vs. 8). Such a result is in agreement with what was previously reported [22] and confirms that the PLANTS scores are particularly effective when applied to complexes directly computed by the PLANTS program but suffer when evaluating complexes optimized using different parameters or force-fields. Accordingly, one may understand the increase in the performances of the VEGA-based energies which are mostly computed by using the same parameters used for the minimization of the complex.

### 2.2. Two-Variable Consensus Models

Table 2 shows the EF averages and the corresponding EF enhancements as obtained by two-variable models for the test sets during the per target validation phase. As seen in Table 1, the EF values are subdivided according to the target classes for the three mentioned parts of the analysis. The reported EF values clearly emphasize the beneficial effects exerted by the inclusion of a second docking score in the generated consensus models. On the one hand, Table 2 confirms the efficacy of consensus strategies based on linear combinations of more than one scoring function; on the other hand, the obtained results emphasize the reliability of the here utilized EFO approach to develop these consensus equations. Table 2 shows clear enhancements in all monitored EF values and in all the parts of the study even though with some differences which deserve specific considerations.

As seen for the EF average values, the EF enhancements are higher when considering the top of the rankings and decrease when extending the monitored rankings. Moreover, the EF enhancements significantly vary in the top of the rankings in the three study sections, while they are marginal and almost constant regardless of rescoring and/or minimization when considering larger parts of the rankings. The limited efficacy of the proposed consensus strategy when analyzing larger parts of the ranking can be explained by remembering that the EFO algorithm develops the linear equations by optimizing a cost function which, while also accounting for the distribution of the active compounds in the entire ranking, and is primarily focused on the number of active compounds within the first cluster. Since, the DUD datasets comprise on average ~ 3000 compounds (see Appendix A) and the size of the cluster is always set to 100 (i.e., 3%), one may understand why the here obtained enhancements are roughly focused on EF values as calculated for the Top 1% and even better for the Top 2% and progressively decreases over the Top 5%. Taken together, these results suggest that the cluster size equal to 100 represents a reasonable choice to obtain a satisfactory early recognition and such a size should be better calibrated only for very small or very large datasets.

When comparing the EF averages and the relative increases for the three parts of the study, one may notice the same trends already observed for the one-variable models with the scoring functions obtained by rescoring the minimized complexes which perform markedly better than the primary PLANTS scores. This observation suggests that this consensus strategy cannot upset the reliability of each single scoring function and thus the best models are conceivably generated by linearly combining the best performing scores. 

The limited improvements observed when using only the PLANTS scores can be explained by considering that the generated equations combine the various normalizations of the sole primary PLP score and thus the included variables, even when surpass the applied cross-correlation filter, comprise rather redundant information. In contrast, the other two parts of the study are based on conceptually diverse docking scores and thus clearly benefit from their linear combinations as seen in Table 2.

The analysis of the results obtained for the different target classes reveals trends very similar to those already seen for the one-variable models. Kinases remain the most challenging targets even when using consensus equations, while metalloenzymes and serine proteases show the highest EF averages. More importantly, the EF differences between one- and two-variable equations as computed for each target class (see Appendix A) reveal that the most challenging targets benefit from the largest enhancing effect by using these linear combinations as exemplified by the kinases EF values. Such an observation emphasizes that the here proposed consensus approach can be particularly fruitful to improve the results when simulating particularly challenging targets which would produce unsatisfactory results. Finally, Appendix A emphasizes the key role of complex minimization. Indeed and in the well performing targets (such as Metalloenzymes), the consensus strategy reveals significant effects only when using the minimized complexes.

Appendix A report the best two-variable equations and the corresponding enrichment factors for each target in the three mentioned analyses. With regard to PLANTS-based models (Appendix A), the reported equations confirm that the non-normalized PLP score plays a limited role and appears only in 12 models; most equations involve pairs of normalized values combining scores normalized per molecular size and per number of contacts. The analysis of two-variable equations as generated by rescoring without complex minimization (Appendix A) confirms the key role of PLANTS scores which appear in 27 out of 40 equations followed by VEGA-based energy components included in 16 models. AMMP-based energies and X-Score values show a similar occurrence since they are involved in 10 and 13 models, respectively. The two-variable models generated by rescoring after complex optimization reveal trends in line with those already observed for one-variable results. In detail, the VEGA-based scores show a remarkable increase in their role since they are involved in 28 models at the expense of PLANTS scores included only in 15 equations. AMMP-based energies and X-Score values roughly retain the same relevance since they are included in 13 and 11 equations, respectively. 

By considering the encouraging results provided by the VEGA-based scores, an additional study involved the development of two-variable equations by including only the VEGA-based scores computed after complex minimization. The study has the primary objective to evaluate in depth the recently proposed scores based on both the Molecular Lipophilicity Potentials [23] (the so-called MLP Interaction score [24]) and on the number of surrounding residues [25] (the Contacts score). These scores proved successful in specific correlative studies (see e.g., ref. [26]), but were never evaluated in extended VS campaigns. Table 2 summarizes the obtained results and emphasizes that, on average, the VEGA-based energy score compares well with PLANTS scores providing comparable EF averages. Interestingly, the analysis of the EF averages for the target classes reveals that these VEGA-based scores do not follow the trends observed in the previous analyses but proved particularly effective for some challenging targets as exemplified by the Nuclear hormone receptors. Appendix A compiles the enrichment factors and the best two-variable equations for each target as generated by using VEGA-based scores and bears witness for the reliability of both MLP Interaction and Contacts scores which are included in 19 and 16 models, respectively.

### 2.3. Three-Variable Consensus Models

As a preamble and due to the above-mentioned redundant information encoded by the various normalizations of the primary PLP score, the three-variable equations were developed including only the docking scores as generated by rescoring calculations with and without complex minimization. 

Table 3 lists the EF averages and the corresponding EF enhancements as obtained by three-variable models during the validation procedures and subdivided according to the target classes for the two remaining parts of the study. Table 3 shows that the inclusion of a third docking score induces EF enhancements which are on average in line with the trends observed in the two-variable equations, since the monitored beneficial effects are more marked when using the minimized complexes and when focusing on the top of the rankings. On average, the inclusion of a third variable induces a more limited enhancement compared to that exerted by the inclusion of a second variable as seen in Table 2. Such an observation suggests that equations combining two or three scores should represent a reasonable compromise to balance performances and computational costs while more complex equations would require a computational time which is not counterbalanced by corresponding enhancements. Table 3 further confirms the efficacy of the linear combinations of docking scores as a consensus strategy to improve the performances of the VS simulations and underlines the fruitfulness of the EFO approach to generate these consensus equations.

The analysis of the EF averages as subdivided per target classes confirms the trends already observed in the previous sections with kinases that remain the most challenging targets. The EF enhancements of each target class (Appendix A) also confirm that the challenging targets are those which mostly benefit from such a consensus strategy. As a matter of fact, kinases represent the most convincing demonstration of the efficacy of the here proposed consensus strategy since their EF average as computed for the Top 1% ranges from 6.4 when using the single best performing PLANTS score to 23.5 when linearly combining scoring functions as computed by using minimized complexes with a more than three-fold overall increase. 

Appendix A show the enrichment factors and the best three-variable equations for each target in the two mentioned analyses. The equations generated by using non-minimized complexes further emphasize the reliability of PLANTS and VEGA scores which appear in 31 and 23 models, respectively, while AMMP energies and the X-Score show a more limited and similar relevance with 13 and 12 occurrences, respectively. When rescoring minimized complexes, one may notice the same trends already observed in the previous analyses, the VEGA-based energy values increasing their role (found in 34 models) at the expense of the PLANTS scores (included in 22 models only). Remarkably, the three-variable equations further confirm the reliability of the MLP Interaction Scores and Contacts scores which are included in 14 and 10 models, respectively.

Finally, Appendix A compares the overall EF means as collected in Table 2 and Table 3 with the corresponding EF values as obtained by applying the generated models to the entire DUD datasets and shows differences between them which are in general limited and progressively decrease when considering wider ranking portions to vanish after the Top 10%. The lack of model performance mismatch emphasizes the reliability of the proposed method which generates robust models even when analyzing very unbalanced datasets avoiding typical problems such as model overfitting or unrepresentative data sampling. By considering these encouraging results, the following comparative analysis will involve the AUC values as computed on the entire DUD datasets in order to offer a more comprehensive comparison with the results provided by other VS approaches which usually does not involve validation procedures apart from those based on machine learning methods.

### 2.4. Comparison with Already Published Studies

As mentioned in the Introduction, this study was based on the DUD datasets since a huge number of even very recent benchmarking studies utilizes them to assess the reliability of various VS techniques; these studies allow the here proposed consensus strategy to be extensively compared with other highly performing VS approaches. Firstly, attention was focused on a set of very recent papers which apply innovative VS methodologies, such as interaction fingerprints [27], machine learning based scoring [26] and ligand-based screening under shape constraints [25]. Along with the novelty of the implemented approach, these studies were chosen because they comprised all available DUD datasets thus allowing a precise comparison with the here reported results. Finally, the comparative analyses were extended to a set of well-established VS techniques to offer a more general evaluation of the proposed consensus approach.

Table 4 compares the AUC values of the here obtained best performing models as computed considering the entire DUD datasets with those reported by the three above-mentioned reference studies. Even though the EFO algorithm yields the best performances when considering the enrichments in the top of the ranking (see above), the reported AUC values reveal that the proposed consensus strategy induces clear enhancements even when considering the entire ranking. Table 4 indicates that the linear combinations of three variables induce an enhancement average of 14% in the AUC values compared to the corresponding best one-variable models. In detail, the proposed consensus strategy provides marginal or nil enhancing effects (i.e., ΔAUC < 3%) only in 10 datasets, while in half of the cases (19 out of 40) the obtained enhancement is greater than 10% with the maximum effects observed for the COMT dataset with an increase of 90%.

The reached AUC values compare very well with those reported in the other three considered studies. Notably, even the models including only one variable reveal performances comparable with those provided by the PADIF and mRAISE methods, while the inclusion of two or three variables yields models with performances clearly better than the mentioned methods and comparable to (if not slightly better than) those produced by machine learning (ML) techniques.

When considering the various classes of targets, the here proposed consensus approach yields the best performances in all considered classes apart from kinases where the ML study reports markedly better results. This finding is in agreement with the already discussed difficulty in screening the kinase datasets and suggests the opportunity of testing diverse approaches in order to find the best performing strategy. Thus, the here reported method affords the best model in 19 out of 40 datasets, the same result is reached by the ML approach, while in 2 cases the best results are provided by the mRAISE method. 

Finally, Figure 1 graphically compares the performances of the reported consensus models including one, two and three variables with those reported by an extended set of benchmarking analyses including the three above considered studies. Even though this comparison is far from being exhaustive and many studies based on the DUD datasets are not here considered, Figure 1 clearly confirms the remarkable reliability and the encouraging performances provided by the presented consensus strategy based on the EFO algorithm. In detail, the generated linear combinations with two and three scores yield highly performing models which are placed in the top (with three variables) and third (with two variables) position in the performance ranking. Interestingly, the still satisfactory performances reached when utilizing one-variable models emphasize the beneficial role exerted by the rescoring procedures, which prove to be a successful method to optimize the reliability of a single docking simulation.

## 3. Materials and Methods

### 3.1. Preparation of DUD Datasets

As mentioned in the Introduction, the present study involved all the 40 datasets included in the original version of DUD (see Appendix A, http://dud.docking.org/). All utilized molecules (protein targets and ligand datasets) were downloaded from the DUD website, while the ligand structures were utilized as they were, the protein structures underwent a preliminary set-up before docking simulations. In detail, the resolved complexes were assembled and completed by adding the non-polar hydrogen atoms using VEGA (the polar hydrogens are already included in the retrieved files) [21]. The so completed protein structures underwent energy minimizations by keeping fixed the backbone atoms to preserve the resolved folding. After deleting the bound ligands, the optimized protein structure underwent docking simulations as detailed below.

### 3.2. Docking Simulations

All docking simulations were performed by using PLANTS [29] which generates reliable poses by *MAX-MIN* ant colony optimization (ACO) [30]. In detail, the searches were focused on a 12.0 Å radius sphere around the bound ligand, a sphere size which is large enough to encompass the entire binding site in all simulated proteins. Docking simulations generated one pose per ligand which was scored by the PLP score function with a speed equal to 2, a set of parameters which proved successful in a recent comparative study. The computed poses underwent rescoring calculations by using the recently proposed ReScore+ tool with and without complex minimization which was performed by keeping all atoms fixed apart from those included in a 10 Å radius sphere around the bound ligand [31]. All mentioned minimizations were performed using NAMD [32], applying the conjugate gradient algorithm with the Gasteiger’s atomic charges and the CHARMM force field [33]. In detail and for each generated complex, the following scoring functions were computed: the Lennard-Jones term of the CHARMM force field, the Lennard-Jones term of the CVFF force field, the electrostatic term as computed with dielectric constant set to 1, the electrostatic term with a distance dependent dielectric constant, the MLP Interaction scores [34], the Lennard-Jones term of the SP4 force field as implemented in the AMMP program [19], the number of contacts [22], the three scoring functions implemented by PLANTS (i.e., ChemPLP, PLP and PLP95) [29], and the X-Score function [20]. 

### 3.3. Generation and Validation of Predictive Models

As discussed in the Introduction, the primary objective of the study involves the assessment of suitably optimized linear combinations of different scoring functions as a consensus strategy for evaluating virtual screening campaigns. The selected linear equations were computed by using the EFO classification algorithm, an approach recently proposed to analyze unbalanced datasets and which can find in virtual screening campaigns a fruitful applicative field [17]. Briefly, the EFO algorithm generates linear combinations of docking scores by an exhaustive search, involving both random selections and optimization procedures. The so-generated consensus models are ranked and selected according to a cost function based on both the enrichment factor analysis and the distribution of active molecule within the entire dataset as encoded by asymmetry-based parameters. Since a calibration study of the key parameters of the EFO algorithm was already performed, here the analyses were carried out by adopting the following conditions: (a) cluster size = 100; (b) interrelated scores are discarded when their VIF >5; (c) ineffective scores are discarded when their single enrichment factor as computed on the top 5% is <2.0; (d) cycles of random sampling performed to generate each starting model = 12; (e) optimization procedures with iterations = 5000 and RMS = 0.001. The consensus equations were generated by including two or three docking scores as computed with and without post-docking minimizations. As an aside and when generating models with only one variable, the EFO algorithm can be also utilized to automatically find the best performing score from among a set of computed scores.

All generated models with two and three variables were evaluated and screened by a per target validation using the procedure implemented by default in the EFO algorithm. Indeed and as previously detailed, the EFO algorithm automatically subdivides the input dataset in training (80%) and test (20%) sets, develops the models using the training set and validates by applying them to the training set. Moreover and to minimize the randomness, this validation task is repeated five times and the models are evaluated and prioritized by considering their average performances as obtained during the validation phase. 

## 4. Conclusions

The study describes the exploitation of linear combinations of more than one docking score as a consensus strategy to enhance the reliability of docking simulations in virtual screening campaigns. Indeed and even though many consensus approaches have been proposed in the last few years, including protocols based on machine learning techniques, the (seemingly simple) linear combinations of scoring functions have received little attention in this field. Here, the consensus linear equations are developed by using the EFO classification method which was recently proposed to analyze highly unbalanced datasets and which can find really fruitful applications for evaluating virtual screening campaigns.

Clearly, an extensive assessment of the performances offered by such a consensus strategy would require further analyses considering (a) different and more challenging datasets, (b) diverse docking programs, (c) additional scoring functions also including molecular descriptors, and (d) more than one computed pose per ligand to be treated in the framework of the recently proposed binding space. That being said, the above-described results provide a convincing confirmation of the remarkable performances of the proposed consensus strategy and allow some general rules to be derived: (1) the linear combination of two or three docking scores represents a satisfactory balance between performances and calculation costs; (2) while clearly depending on the reliability of the computed poses, rescoring calculations appear to be a powerful and straightforward strategy to enhance the performances of a docking simulation; (3) post-docking complex minimization exerts marked beneficial roles especially when considering non-primary scoring functions. As an aside, the study comprises an extensive and successful validation of the reliability of the MLP iteraction scores and the score based on the number of contacts in the field of virtual screening simulations.

## Figures and Tables

**Figure 1 ijms-20-02060-f001:**
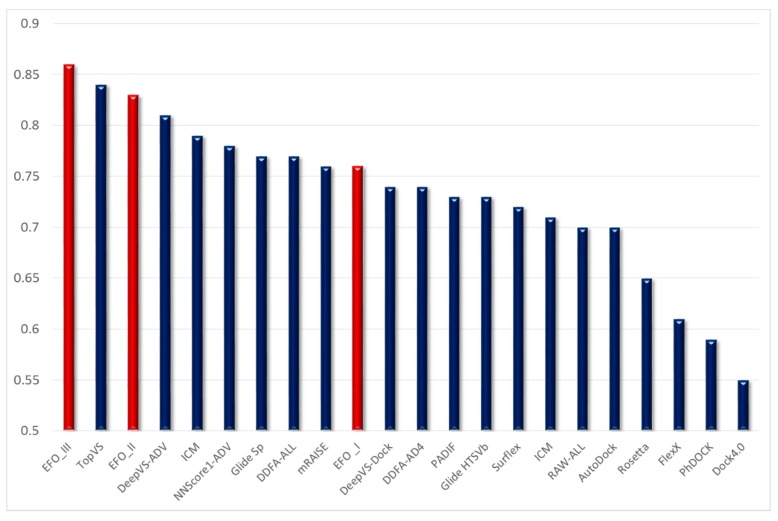
Graphical comparison of the AUC averages as obtained by the here described consensus strategy with those published for a representative set of VS methods (the AUC values were taken from refs. [21,22,24] and references cited therein).

**Table 1 ijms-20-02060-t001:** Enrichment factor averages for the one-variable equations as generated by the EFO algorithm.

Target Class	Score	EF 1%	EF 2%	EF 5%	EF 10%	EF 20%
Kinases	PLANTS	6.4	4.3	2.8	1.9	1.5
Metalloenzymes		17.7	13.6	7.2	4.8	3.0
Nuclear hormone receptors		17.5	12.3	7.7	5.3	3.3
Serine Proteases		14.6	13.3	8.5	5.7	3.5
Other enzymes		17.1	12.2	7.2	4.8	3.0
**Overall mean**		**14.6**	**10.7**	**6.4**	**4.3**	**2.7**
Kinases	Rescore without minimization	15.1	10.5	5.9	3.6	2.2
Metalloenzymes		27.5	16.2	7.3	4.2	2.5
Nuclear hormone receptors		22.2	16.8	10.1	6.4	3.7
Serine Proteases		26.1	18.0	10.9	7.1	4.1
Other enzymes		21.7	15.5	8.2	5.2	3.2
**Overall mean**		**21.4**	**14.9**	**8.2**	**5.1**	**3.1**
Kinases	Rescore with minimization	18.4	12.3	7.1	4.4	2.6
Metalloenzymes		26.3	16.0	7.4	4.2	2.5
Nuclear hormone receptors		23.6	17.9	10.3	6.2	3.6
Serine Proteases		27.1	19.7	13.6	8.0	4.4
Other enzymes		24.4	17.1	10.2	6.0	3.5
**Overall mean**		**23.4**	**16.3**	**9.5**	**5.7**	**3.3**

**Table 2 ijms-20-02060-t002:** Enrichment factor averages for the two-variable equations as generated by the EFO algorithm.

Target Class	Score	EF 1%	EF 2%	EF 5%	EF 10%	EF 20%
Kinases	PLANTS	9.0	6.4	4.0	3.0	2.0
Metalloenzymes		14.4	10.3	6.3	4.3	3.0
Nuclear hormone receptors		22.9	16.4	9.0	5.7	3.5
Serine Proteases		16.5	14.1	8.3	5.1	3.2
Other enzymes		12.6	9.1	5.8	4.1	2.7
**Overall mean**		**15.1**	**11.3**	**6.7**	**4.4**	**2.9**
**Two vs. one variable**		**+0.5**	**+0.6**	**+0.3**	**+0.1**	**+0.2**
Kinases	Rescore without minimization	15.3	10.6	5.9	3.6	2.1
Metalloenzymes		24.4	18.3	10.0	6.3	4.0
Nuclear hormone receptors		24.4	21.1	12.2	7.4	4.1
Serine Proteases		27.3	19.1	11.9	7.0	4.2
Other enzymes		24.2	17.0	9.6	6.1	3.5
**Overall mean**		**23.1**	**17.2**	**9.9**	**5.7**	**3.6**
**Two vs. one variable**		**+1.6**	**+2.3**	**+1.7**	**+0.6**	**+0.5**
Kinases	Rescore with minimization	23.5	16.5	8.7	5.1	2.8
Metalloenzymes		29.3	20.2	10.8	5.9	3.7
Nuclear hormone receptors		25.4	20.9	12.2	7.0	3.9
Serine Proteases		31.7	26.9	15.2	8.1	4.4
Other enzymes		24.7	18.8	10.8	6.2	3.5
**Overall mean**		**26.9**	**20.7**	**11.5**	**6.5**	**3.7**
**Two vs. one variable**		**+3.5**	**+4.4**	**+2.0**	**+0.8**	**+0.4**
Kinases	VEGA with minimization	12.5	9.1	8.9	5.8	3.4
Metalloenzymes		12.0	12.4	9.6	6.4	3.6
Nuclear hormone receptors		20.4	18.2	10.8	6.8	3.7
Serine Proteases		18.6	15.9	10.1	5.9	3.3
Other enzymes		13.9	11.8	8.2	5.4	3.4
**Overall mean**		**15.5**	**13.5**	**9.5**	**6.1**	**3.5**

**Table 3 ijms-20-02060-t003:** Enrichment factor averages for the three-variable equations as generated by the EFO algorithm.

Target Class	Score	EF 1%	EF 2%	EF 5%	EF 10%	EF 20%
Kinases	Rescore without minimization	21.8	15.3	7.7	4.4	2.7
Metalloenzymes		19.3	12.0	8.1	5.0	3.2
Nuclear hormone receptors		30.2	21.5	12.3	7.5	4.5
Serine Proteases		29.9	23.3	14.4	8.1	4.9
Other enzymes		24.2	17.9	9.6	5.8	3.3
**Overall mean**		**25.1**	**18.0**	**10.4**	**6.2**	**3.7**
**Three vs. one variable**		**+3.7**	**+3.1**	**+2.2**	**+1.1**	**+0.6**
**Three vs. two variables**		**+2**	**+0.8**	**+0.5**	**+0.5**	**+0.1**
Kinases	Rescore with minimization	20.5	15.2	8.2	5.0	3.0
Metalloenzymes		26.4	21.1	13.2	7.4	4.0
Nuclear hormone receptors		34.4	28.5	16.4	9.3	4.9
Serine Proteases		33.0	26.8	15.8	8.4	4.3
Other enzymes		26.1	20.4	11.8	7.0	3.9
**Overall mean**		**28.1**	**22.4**	**13.1**	**7.4**	**4.0**
**Three vs. one variable**		**+4.7**	**+6.1**	**+3.6**	**+1.7**	**+0.7**
**Three vs. two variables**		**+1.2**	**+1.7**	**+1.6**	**+0.9**	**+0.3**

**Table 4 ijms-20-02060-t004:** Comparison of the AUC averages as generated by the EFO algorithm with one (EFO I), two (EFO II) and three (EFO III) variables with those reported in the three reference studies based on interaction fingerprints (PADIF [7]), pharmacophore mapping (mRAISE [28]) and machine learning-based scoring (ML [26]). ΔAUC compares the AUC averages as obtained with one and three variables (EFO I vs. EFO III) and thus encodes the overall enhancement exerted by the here described consensus strategy. Underlined values indicate the best results.

DUD Name	Protein class	EFO I	EFO II	EFO III	ΔAUC	PADIF	mRAISE	ML
CDK2	Kinases	0.78	0.82	0.82	0.04	0.81	0.67	**0.88**
EGFr	Kinases	0.67	0.68	0.70	0.03	0.91	**0.96**	0.95
FGFr1	Kinases	0.86	0.88	**0.97**	0.11	0.49	0.54	0.95
HSP90	Kinases	0.71	0.72	0.72	0.01	0.60	0.80	**0.93**
P38 MAP	Kinases	0.62	0.77	0.77	0.15	0.68	0.34	**0.94**
PDGFrb	Kinases	0.53	0.57	0.64	0.11	n.a.	0.35	**0.97**
SRC	Kinases	0.62	0.72	0.84	0.22	0.70	0.45	**0.98**
TK	Kinases	0.66	0.96	**0.96**	0.30	0.71	0.88	0.65
VEGFr2	Kinases	0.66	0.72	0.72	0.06	0.60	0.44	**0.96**
**Mean**	**Kinases**	**0.68**	**0.76**	**0.79**	**0.11**	**0.69**	**0.60**	**0.91**
ACE	Metalloenzymes	0.68	0.74	0.77	0.09	0.46	0.91	**0.81**
ADA	Metalloenzymes	0.79	0.91	**0.91**	0.12	0.90	0.73	0.90
COMT	Metalloenzymes	0.52	0.74	**0.99**	0.47	0.63	0.85	0.73
PDE5	Metalloenzymes	0.74	0.82	0.79	0.05	0.70	0.61	**0.86**
**Mean**	**Metalloenzymes**	**0.68**	**0.80**	**0.87**	**0.18**	**0.67**	**0.78**	**0.83**
AR	NHR	0.69	0.78	0.83	0.14	0.61	0.89	**0.90**
ER agonist	NHR	0.79	0.81	**0.81**	0.02	0.83	0.94	**0.81**
ER antagonist	NHR	0.87	0.86	**0.97**	0.10	0.93	0.92	0.83
GR	NHR	0.75	0.79	0.80	0.05	0.47	0.67	**0.84**
MR	NHR	0.77	0.94	**0.97**	0.20	0.81	0.85	0.87
PPAR	NHR	0.79	0.85	0.85	0.06	0.50	**0.96**	0.72
PR	NHR	0.73	0.77	0.79	0.06	0.72	0.71	**0.91**
RXRa	NHR	0.99	0.99	**0.99**	0.00	0.93	0.90	0.83
**Mean**	**NHR**	**0.80**	**0.85**	**0.88**	**0.08**	**0.73**	**0.86**	**0.84**
FXa	Serine proteases	0.85	0.88	0.88	0.03	0.67	0.71	**0.89**
Thrombin	Serine proteases	0.86	0.98	**0.99**	0.13	0.83	0.68	0.79
Trypsin	Serine proteases	0.97	0.98	**0.99**	0.02	0.95	0.68	0.78
**Mean**	**Serine proteases**	**0.89**	**0.95**	**0.95**	**0.06**	**0.82**	**0.69**	**0.82**
AChE	Other enzymes	0.71	0.71	**0.77**	0.06	0.65	0.75	0.65
ALR2	Other enzymes	0.66	**0.70**	0.68	0.02	0.54	0.61	0.68
AmpC	Other enzymes	0.62	0.91	**0.93**	0.31	0.53	0.91	0.58
COX-1	Other enzymes	0.66	0.68	0.72	0.06	0.39	0.59	**0.86**
COX-2	Other enzymes	0.93	0.92	0.92	-0.01	0.83	0.94	**0.97**
GPB	Other enzymes	0.91	0.92	**0.93**	0.02	0.88	0.92	0.66
HIVPR	Other enzymes	0.81	0.82	0.82	0.01	0.56	0.65	**0.91**
HIVRT	Other enzymes	0.70	0.75	0.81	0.11	0.56	0.64	**0.88**
HMGR	Other enzymes	0.64	0.77	0.71	0.07	0.87	0.95	**0.96**
InhA	Other enzymes	0.64	0.89	0.89	0.25	0.77	0.58	**0.95**
NA	Other enzymes	0.88	0.89	**0.89**	0.01	0.93	0.99	0.87
PARP	Other enzymes	0.86	0.96	**0.96**	0.10	0.74	0.63	0.71
PNP	Other enzymes	0.77	0.92	**0.94**	0.17	0.78	0.99	0.89
SAHH	Other enzymes	0.94	0.97	**0.98**	0.04	0.97	0.98	0.84
DHFR	Other enzymes	0.99	0.99	**0.99**	0.00	0.91	0.99	0.96
GART	Other enzymes	0.96	0.99	**0.99**	0.03	0.96	0.95	0.48
**Mean**	**Other enzymes**	**0.79**	**0.86**	**0.87**	**0.08**	**0.74**	**0.82**	**0.80**
**Overall mean**		**0.76**	**0.84**	**0.86**	**0.10**	**0.73**	**0.76**	**0.84**

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
