# Peer review of "Rescoring and Linearly Combining: A Highly Effective Consensus Strategy for Virtual Screening Campaigns"

_ijms, 2019, doi:10.3390/ijms20092060_

Round 1
Reviewer 1 Report
The paper is an excellent and scientifically relevant work. The proposed consensus strategy to obtain good enrichment factors appears effective and reliable. The whole strategy is well explained and the results are consistent with the final conclusions.
It should be published in the present form, only with minor revisons.
Minor revisions
Rows 70 – 76. In my opinion, some more information should be given about the EFO approach, such avoiding the need to read also the quoted paper [15].
Further comments
Row 106. The first comments on the Tables of the supplementary material: add “(see Supplementary material)”
Table 3. The numbers of the last three rows should be in bold characters.
Row 313. A reference to the ant colony optimization (ACO) should be given.
Author Response
Response to reviewer 1 comments
The paper is an excellent and scientifically relevant work. The proposed consensus strategy to obtain good enrichment factors appears effective and reliable. The whole strategy is well explained and the results are consistent with the final conclusions.
It should be published in the present form, only with minor revisions.
Minor revisions
Rows 70 – 76. In my opinion, some more information should be given about the EFO approach, such avoiding the need to read also the quoted paper [15].
Thank you very much for the valuable comment. More information about the EFO algorithm has been added (Rows 74-80).
Further comments
Row 106. The first comments on the Tables of the supplementary material: add “(see Supplementary material)”
Row 106 was completed according the reviewer advice.
Table 3. The numbers of the last three rows should be in bold characters.
The Table 3 was modified introducing bold characters.
Row 313. A reference to the ant colony optimization (ACO) should be given.
Since PLANTS implements MAX-MIN ACO algorithm, the sentence was completed with this information and the related reference was added (#27).
Reviewer 2 Report
The manuscript authored by Pedretti et al is very interesting. the paper is well written and clearly presented. The authors carried out a significant computational investigation in order to evaluate the performance of a consensus strategy that can be improved the ratio for finding hit compounds in virtual screening procedure. I strongly support the publication of the paper after few corrections mainly regarding the lack of recently research papers regarding the use of EF for assessing the performance of the in silico models.
Please add these two references in the introduction part
G Chemi et al Computational tool for fast in silico evaluation of hERG K+ channel affinity Frontiers in chemistry 2017, 5, 7 https://www.frontiersin.org/articles/10.3389/fchem.2017.00007/full
L Zaccagnini et al Identification of novel fluorescent probes preventing PrPSc replication in prion diseases European journal of medicinal chemistry 2017, 127, 859-873 https://www.sciencedirect.com/science/article/pii/S0223523416309357
In addition I kindly ask to clarify why the authors use DUD instead of DUD-E that represent the enhanced version of the database use in this paper
Author Response
Response to reviewer 2 comments
The manuscript authored by Pedretti et al is very interesting. the paper is well written and clearly presented. The authors carried out a significant computational investigation in order to evaluate the performance of a consensus strategy that can be improved the ratio for finding hit compounds in virtual screening procedure. I strongly support the publication of the paper after few corrections mainly regarding the lack of recently research papers regarding the use of EF for assessing the performance of the in silico models.
Please add these two references in the introduction part
G Chemi et al Computational tool for fast in silico evaluation of hERG K+ channel affinity Frontiers in chemistry 2017, 5, 7 https://www.frontiersin.org/articles/10.3389/fchem.2017.00007/full
L Zaccagnini et al Identification of novel fluorescent probes preventing PrPSc replication in prion diseases European journal of medicinal chemistry 2017, 127, 859-873 https://www.sciencedirect.com/science/article/pii/S0223523416309357
Both references have been added to the introduction.
In addition I kindly ask to clarify why the authors use DUD instead of DUD-E that represent the enhanced version of the database use in this paper.
We preferred to perform our VS analysis on DUD instead of its enhanced version DUD-E, because in literature there are many studies based on the older dataset, making possible extensive comparisons.
The sentence at row 81 was modified in order to better explain the reason of this choice.